# The Optimal Concentration of Formaldehyde is Key to Stabilizing the Pre-Fusion Conformation of Respiratory Syncytial Virus Fusion Protein

**DOI:** 10.3390/v11070628

**Published:** 2019-07-08

**Authors:** Wei Zhang, Lu-Jing Zhang, Lu-Ting Zhan, Min Zhao, Guang-Hua Wu, Jun-Yu Si, Li Chen, Xue Lin, Yong-Peng Sun, Min Lin, Chao Yu, Mu-Jin Fang, Ying-Bin Wang, Zi-Zheng Zheng, Ning-Shao Xia

**Affiliations:** 1National Institute of Diagnostics and Vaccine Development in Infectious Diseases, School of Life Sciences, Xiamen University, Xiamen, Fujian 361005, China; 2State Key Laboratory of Molecular Vaccinology and Molecular Diagnostics, School of Public Health, Xiamen University, Xiamen, Fujian 361005, China

**Keywords:** respiratory syncytial virus, fusion protein, pre-fusion conformation, antigenic sites, fixatives, formaldehyde, stabilization, protection, pulmonary inflammation

## Abstract

Background: To date, there is no licensed vaccine available to prevent respiratory syncytial virus (RSV) infection. The valuable pre-fusion conformation of the fusion protein (pre-F) is prone to lose high neutralizing antigenic sites. The goals of this study were to stabilize pre-F protein by fixatives and try to find the possibility of developing an inactivated RSV vaccine. Methods: The screen of the optimal fixative condition was performed with flow cytometry. BALB/c mice were immunized intramuscularly with different immunogens. The serum neutralizing antibody titers of immunized mice were determined by neutralization assay. The protection and safety of these immunogens were assessed. Results: Fixation in an optimal concentration of formaldehyde (0.0244%–0.0977%) or paraformaldehyde (0.0625%–1%) was able to stabilize pre-F. Additionally, BALB/c mice inoculated with optimally stabilized pre-F protein (opti-fixed) induced a higher anti-RSV neutralization (9.7 log_2_, mean value of dilution rate) than those inoculated with unstable (unfixed, 8.91 log_2_, *p* < 0.01) or excessively fixed (exce-fixed, 7.28 log_2_, *p* < 0.01) pre-F protein. Furthermore, the opti-fixed immunogen did not induce enhanced RSV disease. Conclusions: Only the proper concentration of fixatives could stabilize pre-F and the optimal formaldehyde condition provides a potential reference for development of an inactivated RSV vaccine.

## 1. Introduction

Respiratory syncytial virus (RSV) induces a serious lower respiratory tract infection [1] and it is considered to be associated with wheezing in children (<3 years old) [2,3], especially in preterm infants [4]. It is one of the primary causes of virus-induced infant hospitalization [5,6] and it is related to substantial mortality in both developed and developing countries [5,7]. Palivizumab, which targets the fusion protein (F protein) of RSV, is the only licensed monoclonal antibody (mAb) for the prevention of RSV infection in high-risk infants, but its use is not widespread due to its high cost. However, despite numerous vaccines currently undergoing clinical trials or preclinical study, there is still no licensed vaccine for RSV available.

Formalin is a commonly used vaccine-inactivating agent. Formaldehyde can react with amino groups at N-terminal of amino acids, as well as at the side chains of arginine, cysteine, histidine and lysine residues to stabilize and inactivate proteins [8]. In the 1960s, the formalin-inactivated RSV (FI-RSV, lot 100) candidate vaccine showed no protection, but instead caused enhanced RSV disease during phase II clinical trials. FI-RSV produced a 16-fold higher hospitalization rate than the control vaccine and resulted in the death of two infants [9]. Studies over the past few decades have provided important information indicating that FI-RSV does not induce high levels of RSV-specific neutralizing and fusion-inhibiting antibodies [10,11]. It was also reported that the carbonyl group on formaldehyde-treated vaccine antigens boost the th2-biased response and enhance pulmonary disease in mice [12]. As such, the FI-RSV vaccine did not inhibit virus infection or suppress virus replication, but exacerbated pulmonary pathology [10,13]. Recent studies have shown that the pre-F protein did not exist on FI-RSV virions because it was lost during the production of FI-RSV [14]. This raises two important questions: 1) What factors determine the loss of pre-F protein during the preparation of FI-RSV; and 2) what methods could be used to stabilize the pre-F protein while inactivating RSV? Therefore, the purpose of this study was to examine the stability of the pre-F protein on RSV infected cells in various fixatives and applicable conditions. Our results demonstrate the optimal formaldehyde and paraformaldehyde conditions that stabilize the pre-F protein and its functional neutralizing antigenic site. These data also provide a more accurate explanation for the loss of the pre-F protein in FI-RSV—that is, the inactivation concentration of FI-RSV was too low to stabilize the pre-F protein. Moreover, in BALB/c mice, infected cells fixed with the optimal concentration of formaldehyde exhibited a higher efficacy in preventing RSV infection than untreated cells. The results from this study provide insights into the stabilization of the pre-F protein and high neutralizing antigenic sites with fixatives and increases the possibility of developing a new FI-RSV vaccine.

## 2. Materials and Methods

### 2.1. Cell and Virus Culture Conditions and Animal

HEp-2 cells (ATCC, Manassas, VA, USA) were maintained in an Minimum Essential Medium (MEM) (Gibco, Thermo Fisher Scientific, Waltham, MA, USA) supplemented with 10% FBS (Gibco) and 100 U/mL penicillin–streptomycin (Gibco). RSV strain A2 (ATCC) and RSV-A2-mKate (pSynkRSV A2 D46F, kindly donated by Dr. Barney S. Graham, VRC, NIH; this virus is a recombinant based on RSV strain A2 inserting a fluorescent reporter before NS1 protein) were produced in HEp-2 cells. Specific-pathogen free BALB/c mice used in this study were purchased from the SLAC Laboratory Animal Co., Ltd. in Shanghai, China. All of the animal experiments performed in this study were approved by the Institutional Animal Care and Use Committee and Laboratory Animal Management Ethics Committee of Xiamen University (The ethic approval number: XMULAC20170347).

### 2.2. Antibodies

Primary antibodies 5C4 (specific to site Ø on pre-F conformation), 8C2 (binds to both the pre-F and post-F protein conformation) and 9F7 (specific to HEV, used here as negative control) were purified from the seroperitoneum of mice using Protein A Resin (GenScript, Piscataway, NJ, USA). We expressed 1129, AM14, AM22, 101F, MPE8 and Motavizumab using FreeStyle^TM^ 293-F cells (Gibco) via eukaryotic expression plasmid kindly donated by Jason S McLellan (Department of Molecular Biosciences College of Natural Sciences, The University of Texas at Austin, Austin, TX, USA).

### 2.3. Cell Infection and F-Protein Epitope Stabilization 

Cells were infected with RSV strain A2 at a MOI = 3. Twenty-four hours post-infection cells were digested with 0.005% Trypsin-EDTA (Gibco), resuspended in phosphate buffered saline (PBS) buffer and split into sterile EP tubes. Then cells were treated with different fixative conditions. Formaldehyde (Xilong Scientific Co., Ltd., Shanghai, China), methanol (Xilong Scientific Co., Ltd.), beta-propiolactone (SERVA, Heidelberg, Germany), paraformaldehyde (Sigma-Aldrich, St. Louis, MO, USA) and glutaraldehyde (Sigma-Aldrich) were all diluted with PBS according to concentration gradients to generate different fixative conditions. After incubation, the fixatives were removed via centrifugation (200× *g*, 3 min, 25 °C), and then the cells were washed with PBS one time.

### 2.4. Detection of Epitopes on RSV-Infected Cells

The fixative-treated cells were incubated with primary antibodies at a final concentration of 20 ng/μL for 20 min at 25 °C. The cells were then washed with PBS and then incubated with a secondary antibody, either GAM-FITC (goat anti-mouse immunoglobulin fluorescein isothiocyanate conjugate, Sigma-Aldrich) or GAH-FITC (goat anti-human immunoglobulin fluorescein isothiocyanate conjugate, Sigma-Aldrich) at a ratio of 1:2000 for 20 min at 25 °C. Samples were washed once with PBS to remove the free secondary antibodies and then analyzed via flow cytometry (Facsaria III, BD Biosciences, San Jose, CA, USA). Data were collected and then analyzed with FlowJo version 7.6.1 (BD Biosciences).

### 2.5. BALB/c Vaccination and RSV Challenge

SPF BALB/c (six-week-old) mice were grouped and inoculated with different vaccine mimics subcutaneous injection, and then with 1 × 10^7^ cells per mouse on day 0, 10 and 20. Serum samples from each mouse were collected for further analysis. Four weeks after the third inoculation, BALB/c mice were challenged with RSV strain A2 at a dose of 1 × 10^7^ pfu/mouse. Mouse weights were recorded on day 0 and day 7 post-challenge. On day 5, five mice were killed, and the right lungs were obtained. Lung homogenates were made using a gentleMACS Dissociator (Miltenyi Biotec, Bergisch Gladbach, Germany) on program Lung 02. Pulmonary virus titration was determined via plaque assay, and replication of the RSV A2 genome RNA was detected by measuring the target RSV A2 N gene via qPCR on a CFX96 Touch™ Real-Time PCR Detection System (Bio-Rad, Hercules, CA, USA) as previously described [15,16]. The left lungs were kept in 10% formaldehyde and paraffin-embedded. A pathologist who did not have knowledge of the group assignment of individual mice scored lung pathology according to the Cimolai method [17]. In brief, peribronchiolar/bronchial infiltrates (‘A’, the percentage of bronchioles with significant infiltrates), quality of peribronchiolar/bronchial infiltrates (‘B’, degree of infiltration of peribronchiolar inflammatory cells, which can be divided into four grades according the thickness of inflammatory cells), bronchiolar/bronchial luminal exudate (C), perivascular infiltrates (‘D’, the percentage of vascular with significant infiltrates) and parenchymal pneumonia (‘E’, the degree of parenchymal pneumonia can be evaluated by checking the presence of patchy parenchymal infiltrates and confluent parenchymal infiltrates) were evaluated by scanning the whole section of every mice. Then the pathology score was calculated with the following formula: Pathology Score = A + 3 (average B + average C) + D + E. To compare eosinophil infiltration in the lung sections, we evaluated the number of eosinophil stained cells in and around blood vessel walls. We also measured the lengths of the blood vessel walls. The results were expressed as the number of eosinophils per millimeter of vessel wall.

### 2.6. Serum Adsorption and Neutralization Assay

The contribution of anti-pre-f antibodies to serum neutralizing activity can be analyzed by blocking these antibodies with DS-CAV1. In brief, the serums were mixed with purified DS-CAV1 protein (1 μL serum every 1 ug protein) and the mixture is incubated at room temperature for 1 h to allow the antibody to fully bind to DS-CAV1. The neutralization of these mixtures against RSV was then assessed.

The RSV neutralization assay using anti-F monoclonal antibodies or mouse serum after vaccination was performed with HEp-2 cells and RSV-mkate (pSynkRSV A2 D46F) [18]. In brief, cells were seeded in 96-well microplates at density of 30,000 per well and incubated for six hours at 37 °C with 5% CO_2_. Then 10 μL of non-blocked or 20 μL DS-Cav1-blocked serum or mAb (1 μg/μL) were added into 90 μL or 80 μL MEM medium and then serially four-fold diluted. Then 75 μL of the serially diluted samples were mixed with an equal volume of RSV-A2-mKate virus (diluted in advance with MEM at the concentration of 9 × 10^4^ PFU per 50 μL). The mixtures were incubated for one hour at 37 °C with 5% CO_2_. The medium from the cell plate was then removed, and 100 μL was added to the cells for incubation. Twenty-four hours later, the fluorescence intensity was captured with a SpectraMax Paradigm Multi-Mode Microplate Reader (Molecular Devices, LLC, San Jose, CA, USA) at 588 nm excitation and 633 nm emission. The IC50 and inhibition curve were computed by GraphPad Prism version 7.00 (GraphPad Software, San Diego, CA, USA, http://www.graphpad.com).

### 2.7. Serum IgG Isotype Analysis

According to recent reports [19], the isotypes of IgG antibodies in serum were measured by enzyme-linked immunosorbent assay (ELISA). In short, ELISA plates were coated with RSV pre-F protein or RSV G protein in phosphate buffer (pH 7.4) and incubated overnight at 4 °C. Sera were then serially 10-fold diluted and added into the antigen-coated plates and incubated at 37 °C for one hour. After five washes, horseradish peroxidase (HRP)-conjugated goat anti-mouse IgG1 or IgG2 antibody (SouthernBiotech, Birmingham, AL, USA) were added into wells at the dilution of 1:2000 for detecting the different IgG isotypes. Serum antibody titer was expressed as the highest dilution of serum giving an absorbance reading value greater than 0.3.

### 2.8. Plaque Assay

The titer of virus stock or lung homogenate was determined as described previously [15]. Briefly, virus stocks or lung homogenates were serially 10-fold diluted with MEM medium. Then 50 μL of the diluted samples from the dilution of 1 × 10^3^ to 1 × 10^7^ were added to a monolayer of Hep-2 cells (2 × 10^5^ per well, 12-well plate) for one hour at 25 °C, and then transferred into the 37 °C incubator with 5% CO_2_. Four days later, plaques were stained with hematoxylin and eosin (H&E) (Sigma-Aldrich) and counted.

### 2.9. Detection of the F Protein on RSV

The pre-F/Total-F antigenic sites on RSV were detected via Dot-Blot, which was performed on Bio-Dot SF Microfiltration System (Bio-Rad, USA) and ChemiDoc MP imaging system (Bio-Rad, USA) according to the instruction manuals. In brief, samples were loaded onto the nitrocellulose membrane with the prepared microfiltration system apparatus, 100 μL per well. After blocked with 5% (*wt*/*vol*) skim milk for one hour, membranes were respectively incubated with primary antibodies (RSV pre-F mAb 5C4, 2 ng/μL; RSV F mAb 8C2, 0.3 ng/μL the negative control, HEV ORF2 mAb 9F7, 0.3 ng/μL). After three washes, membranes were further incubated with a secondary antibody, goat anti-mouse IgG-FITC (SIGMA, USA) at a dilution of 1:3000 for one hour. Finally, ChemiDoc MP full-wavelength gel imaging system was used for scanning and photographing of membranes at 488 nm excitation.

### 3.10. Statistical Analysis

Statistical analysis (Nonparametric test) of the data was performed using GraphPad Prism version 7.00. A *p* value of <0.05 was considered statistically significant.

## 3. Results

### 3.1. Pre-F Protein-Specific Antigenic Sites on RSV-Infected Cells were Sensitive to Fixative Concentration

We first measured the presence and conformation of F proteins on infected cells after the cells were harvested and stored in PBS at 37 °C (Figure 1A) or 4 °C (Figure 1B). Since we could not easily express eukaryotic antibodies, we mainly selected 5C4 and 8C2 antibodies for the main follow-up experiments. Antibody 5C4, which recognizes pre-F proteins, and antibody 8C2, which recognizes both pre- and post-F proteins, were used to detect pre-F and total-F proteins, respectively. The decline in the 5C4 reaction was obvious during the storage of cells in PBS, especially after one hour, while the reaction with 8C2 was stable. Our results demonstrated that low temperature (4 °C, Figure 1B upper panel) was better at maintaining the 5C4 reaction compared to 37 °C (Figure 1A upper panel). However, cell disruption and agglomeration gradually appeared after 12 h of incubation in PBS, both at 4 °C and 37 °C, and the detection of pre-F and total-F proteins became undetectable. The results indicated that most pre-F protein on the surface of the cells were transformed to post-F within 12 h, which is consistent with recent reports, which indicated instability of the pre-F protein on RSV virions [14].

We further measured the influence of four fixatives on the stability of the F protein. The reaction of cells with 5C4 was significantly higher after treatment within specific concentrations of formaldehyde (0.0244% and 0.0977%, Figure 2A) or paraformaldehyde (0.0625%, 0.25% and 1%, Figure 2B) than that in PBS. Additionally, treatment with methyl alcohol or glutaraldehyde showed no obvious improvement in the reaction of cells with 5C4 (Figure 2C,D). We also noticed that reactivity of the mAbs to F proteins varied with fixative concentration. In low concentrations of fixative, the reactivity of 5C4 (pre-F protein), but not 8C2 (total-F protein), declined, which was consistent with the metastable feature of the pre-F protein. This implied that low concentrations of fixative did not inhibit conformational changes in the pre-F protein. In contrast, high concentrations of fixative reduced the reactivity of both mAbs and should not be considered as the result of pre-F transformation to post-F. Furthermore, prolonging the fixative incubation period induced a significant decrease in high and low fixative concentrations, as well as the optimal fixative concentration. These results were similar to the influence of concentration change on the stability of the F protein (Figure 2). Taken together, our results demonstrate that an optimal concentration of a specific fixative for a certain length of time contributed to the stabilization of the pre-F protein, whereas over-fixation resulted in the destruction of antigenic sites on the F protein. Furthermore, a low concentration of fixative did not increase the stabilization of pre-F proteins. The pre-F conformation showed relative sensitivity to fixative concentrations and duration of fixation.

### 3.2. Optimal Formaldehyde Fixation Stabilized the Major Neutralizing Antigenic Sites of the Pre-F Protein on RSV-Infected Cells

We assessed the reactivity of the 5C4 and 8C2 antibodies to the F proteins on the cells that were pretreated with 0.0061%, 0.0244% or 0.0977% formaldehyde and then stored in PBS for 12 h at 37 °C. As shown in Figure 3A, although the 5C4 reactivity of the 0.0244% formaldehyde group was highest, the reaction showed a slight decline after keeping the cells in PBS for 12 h. The reaction of the 0.0061% formaldehyde-treated group with 5C4 was the lowest and was almost undetectable after 12 h. The 0.0977%-formaldehyde group displayed a sTable 5C4 reaction in 12 h, and the 8C2 reactivity of all three groups was similar and stable. These results suggest that a higher formaldehyde concentration stabilizes the antigenic sites present on the pre-F protein.

The reactivity of antigenic sites on the F protein in formaldehyde-treated cells was further measured. As shown in Figure 3B, the level of neutralizing antigenic sites on the F protein in the 0.0244% formaldehyde-treated group and fresh harvested cells (100% set as positive control) were almost the same. In addition, the 0.0244% formaldehyde pre-treated antigenic sites were relatively stable in PBS at 37 °C, but not all of the antigenic sites on untreated cells were stable. These results further supported that antigenic sites specially presented on the pre-F protein were unstable within 12 h on harvested RSV-infected cells. Taken together, our results demonstrate that 0.0244%–0.0977% formaldehyde was able to stabilize the pre-F protein and the major neutralizing antigenic sites of the F protein, especially the antigenic sites present on the pre-F protein and recognized by high potency neutralizing antibodies, on the surface of infected cells.

### 3.3. Inoculation with RSV-Infected Cells Fixed with the Optimal Concentration Formaldehyde Protects Mice Against High-Titer RSV Challenge

Previous studies have shown that recombinant pre-F protein could stimulate higher levels of serum neutralizing antibodies than post-F protein at the same vaccination dose [20,21,22,23,24]. Therefore, we further compared the anti-RSV immunogenicity of formaldehyde-fixed RSV-infected cells, which would possess different pre-F-specific antigenic site reactivity. These cells were treated with different concentrations of formaldehyde. The groups included: 1) 0% formaldehyde (unfixed group, which was treated with PBS, and as a result, lost the major pre-F protein-specific neutralizing antigenic sites); 2) 0.0244% formaldehyde (opti-fixed group, which was treated with 0.0244% formaldehyde and displayed a stabilized pre-F protein) and 3) 25% formaldehyde (exce-fixed group, which was fixed with an excessive concentration of formaldehyde and lost the major neutralizing antigenic sites on the F protein). 

After four vaccinations without an adjuvant (1 × 10^7^ cells per mouse; intramuscular), we evaluated the G and F-specific antibody responses (Figure 4A–D). Mice produced high-titer IgG1 isotype antibodies targeting G and F, which were comparable among groups. However, mice in opti-fixed group produced higher levels of G and F-specific IgG2a isotype antibodies than mice in unfixed group (anti-F, *p* = 0.0021) and exce-fixed group (anti-F, *p* = 0.0023). This led to a more balanced immune response of Th1/Th2 in mice of opti-fixed group, while mice in the unfixed group and exce-fixed group were more inclined to Th2. The serum anti-RSV neutralization was a mean 9.7 log_2_ in the opti-fixed group, which was higher than both the exce-fixed group (mean of 7.28 log_2_, *p* < 0.01) and unfixed group (8.91 log_2_, *p* < 0.01) after three boosted doses (Figure 4E). We also noted that after adequate incubation with DS-CAV1, serum neutralization activity was almost eliminated in all groups, indicating that anti-pre-f antibody had an important contribution to serum neutralization activity (Figure 4E). These results implied that pre-F protein fixed with 0.0244% formaldehyde remained relatively stable in vivo.

Moreover, the protective efficacy of the fixed immunogens was evaluated by vaccinating mice with formaldehyde-fixed RSV-infected cells and then challenging these mice with a high-titer of RSV-A2 (1 × 10^7^ PFU/mouse, intranasal). The weight of each mouse was recorded on day 0 and 7 post-challenge, and as shown in Figure 5A, all three of the vaccination groups displayed less weight loss compared to the non-vaccinated group. The mice in the opti-fixed group were well protected and exhibited a 1.153% mean weight loss, which was slightly lower than the unfixed group (mean weight loss = 4.883%, *p* = 0.06) and obviously lower than the exce-fixed group (mean weight loss = 6.09%, *p* = 0.02). We also assessed the pulmonary virus load in the right lungs of five mice on day 5 post-challenge via both plaque assay and qPCR assay. The plaque assay showed negative results in all of the vaccinated groups, which might have been caused by the residual anti-RSV neutralizing antibodies in the lung homogenates. The qPCR results of pulmonary RSV RNA copies, which reflected total virus replication, is shown in Figure 5B. The mean value of the opti-fixed group (5.121 ± 0.391 (log_10_)) was 100 to 1000 times lower than the mean values of the unfixed group (7.650 ± 1.379 (log_10_), *p* = 0.008) and exce-fixed group (8.121 ± 0.651 (log_10_), *p* = 0.008). Notably, the RSV RNA copy of the unfixed group was maintained at a high level, which implied weak viral suppression and was consistent with the serum neutralizing activity and weight loss to some degree.

The left lung samples of the mice were stored in 10% formaldehyde solution prior to paraffin embedding and sectioning. Results of lung pathology scoring are shown in Figure 5C and Appendix A. The mock-vaccinated mice developed relatively serious pulmonary inflammation and slight perivascular inflammation, with the appearance of infiltrating cells in the thickened alveolar wall and alveolar space (Appendix A). We evaluated several pathological indexes, including peribronchiolar/bronchial infiltrates, the quality of peribronchiolar/bronchial infiltrates, bronchiolar/bronchial luminal exudate and perivascular infiltrate and parenchymal pneumonia, to calculate the total pathology score, and as expected, mice in the opti-fixed group developed slight pulmonary histopathological change with peribronchovascular cuffing. Mice in the unfixed (*p* = 0.02) and exce-fixed (*p* = 0.03) groups all developed significantly more serious inflammation than the opti-fixed group, with increased white blood cells, including monocytes, neutrophils and lymphocytes, aggregated around the vessels and bronchus. In addition, mice in the unfixed group suffered multi-focal inflammation in the lungs, which impaired the breathing function of the mice. We also observed significantly fewer pulmonary eosinophils in the lungs of the opti-fixed group mice compared with the unfixed group (*p* = 0.0068) and the exce-fixed group (*p* = 0.0023; Figure 5D), which may indicate a more balanced Th1 and Th2 immune response in the lungs of the opti-fixed group mice. However, it should not be overlooked that higher levels of eosinophilic infiltration did occur in the opti-fixed group than in the mock-vac group (*p* = 0.0483) and MC group (*p* = 0.0111), which suggested that the opti-fixed group mice had a slight Th2-shift in Th1/Th2 balance. To sum up, opti-Fixed RSV-infected cells are capable of protecting mice from serious RSV-induced pulmonary pathological changes. Taken together, 0.0244% formaldehyde-treated infected cells were immunogenic and able to induce a protective immune response against the high-titer RSV challenge.

## 4. Discussion

In this study, we demonstrated that RSV-infected cells expressed abundant pre-F protein on the cell surface, which was unstable and prone to undergo a conformational change, and the major pre-F specific antigenic sites were loss within 12 h of storage in PBS. Furthermore, the optimal concentration of cross-linking via chemical fixatives, including formaldehyde and paraformaldehyde, were able to stabilize the metastable pre-F protein on the cell surface of infected cells. In addition, the F protein after trypsin treatment retained the major antigenic sites, suggesting that slight trypsin treatment had a relatively small effect on the F protein.

Formaldehyde and paraformaldehyde stabilizes proteins via forming a mixture of reversible and irreversible cross-linking and CH2 linkages between amino acid residues—the formation of which depends on the exposure time, temperature and fixative concentration [8]. Here, we found that 0.0244%–0.0977% formaldehyde and 0.0625%–1% paraformaldehyde demonstrated stabilization of the conformation of pre-F protein. This suggested that the proper cross-linking probably formed among several secondary structural elements (especially α2, α3, α4 and the β3/β4 hairpin on F1) [25] before pre-F conformational change and loss of the neutralizing antigenic sites. This might suppress dramatic conformational rearrangement of F1, thereby stabilizing the pre-F protein. Moreover, a lower concentration (<0.0244%) of formaldehyde (or paraformaldehyde) might generate too few cross-links to inhibit rearrangement of F1 before conformational change of the pre-F protein. Additionally, high concentrations of formaldehyde (or paraformaldehyde, methyl alcohol or glutaraldehyde) might cause the formation of excessive cross-linkages on the protein surface, potentially resulting in the loss of majority of the antigenic sites.

It is important to note that the immunogens used in this study were made of infected cells, not viruses, but the results in this paper also suggest that an inactivated vaccine based on purified virions is within reach. We also detected the influence of formaldehyde treatment on RSV virions. It was found that a proper concentration range of formaldehyde enabled the long-term display of pre-F-specific antigenic sites on virions, and the effective concentration range shifted slightly to higher concentrations under 12 h of treatment (Appendix A). It is noteworthy to mention that the production process of the FI-RSV vaccine used 0.01% formaldehyde to inactivate RSV for 72 h at 37 °C [10,11,26]. Compared with our optimal fixative condition, the formaldehyde concentration used in the FI-RSV production process was probably too low to stabilize pre-F protein and could be the key reason why FI-RSV lacked pre-F proteins.

Pre-fusion F-specific antibodies were responsible for the major RSV neutralizing activity of sera, which has been observed in humans, mice, and cotton rats [27,28,29]. In particular, site Ø-specific antibodies were the major contributor to the neutralizing activity, but site V and site III-directed neutralizing antibodies also showed potent virus-neutralizing capacity [30]. In accord with previous reports, our results showed that the mice inoculated with the immunogen with more pre-F protein had a higher serum neutralizing capacity, in which anti-pre-F antibodies played an important role, of course, we could not exclude the possible contribution of anti-G or SH specific antibodies. These mice also showed lower viral replication and less pulmonary inflammation after the RSV challenge. It is worth noting that differences in formaldehyde concentrations used to treat these three immunogens may have potential effects on epitopes (including T cell epitopes), which may be a key factor leading to different degrees of biased Th1/Th2 immune responses in mice.

Our results show that optimal usage of formaldehyde could stabilize the antigenic sites of the metastable RSV pre-F protein. Moreover, the stabilized immunogen containing pre-F antigenic sites could protect hosts against the RSV challenge without resulting in serious pulmonary illness. Our study may provide a feasible method to obtain stable pre-F protein with proper molecular or aggregation morphology, and our results provide insights into the development of an inactivated RSV vaccine. Furthermore, development of an inactivated RSV vaccine appears to no longer be an impasse, but a potential favorable treatment option for RSV infection.

## 5. Patents

There is a patent (Application No.: WOCN17118942) resulting from the work reported in this manuscript.

## Figures and Tables

**Figure 1 viruses-11-00628-f001:**
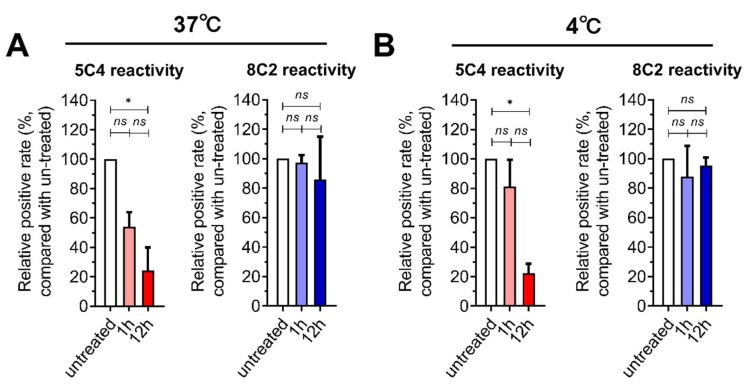
The stability of pre-F expressed on the surface of respiratory syncytial virus (RSV)-infected cells. HEp-2 cells were infected with an RSV A2 strain and were then harvested by trypsin treatment at 24 h post-infection. (**A**,**B**) The cells were re-suspended in PBS at 37° C (A) or 4 °C (B) for one or 12 h. The cells were stained with 5C4 (pre-F) or 8C2 (total F) and a secondary mAb conjugated with FITC and then detected by flow cytometry. The positive gates were set by using uninfected cells as a negative control. The signal from the cells that reacted with the mAbs were measured and compared with untreated RSV-infected cells (the cells were stained and detected immediately after harvest by trypsin, and the positive rate of this group was set as 100%). For statistical assay, a non-parametric statistical test was performed with GraphPad Prism version 7.00. Result means geometric mean ±95% CI. * *p* < 0.05, ** *p* < 0.01, *** *p* < 0.001, *** *p* < 0.0001.

**Figure 2 viruses-11-00628-f002:**
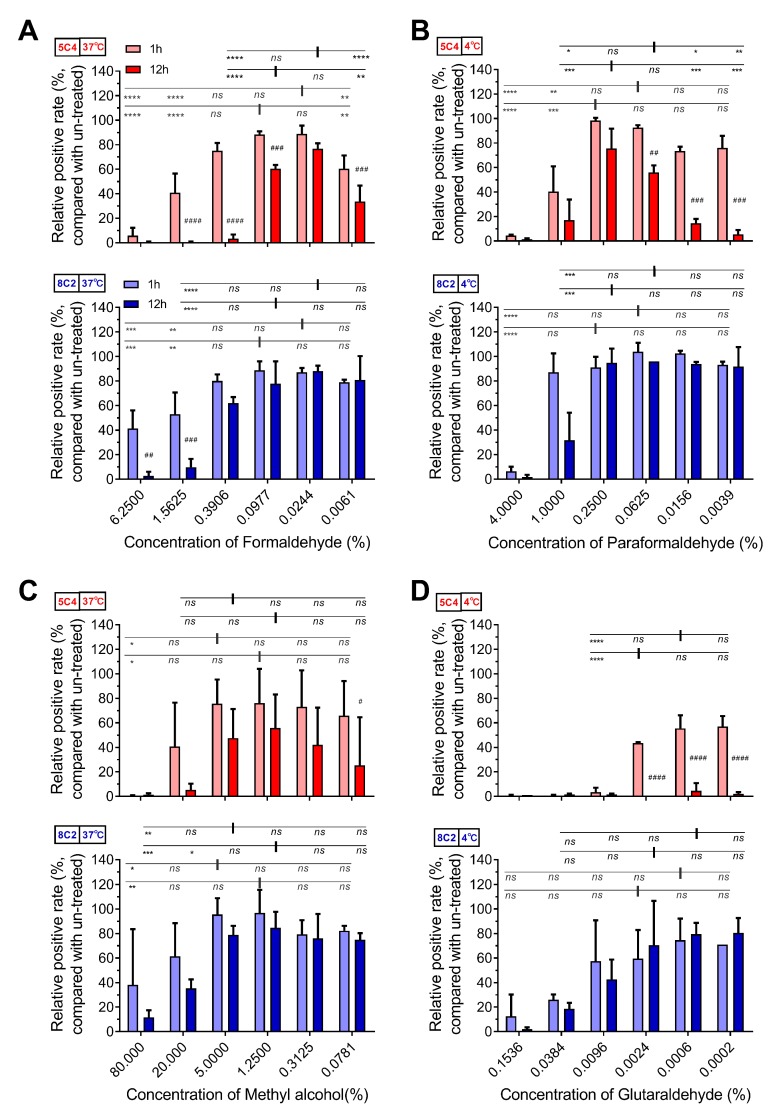
The optimal fixative condition stabilized the F epitopes on RSV-infected cells. HEp-2 cells were infected with RSV A2 strain and were then harvested by trypsin treatment at 24 h post-infection. The cells were treated with serially diluted (**A**) formaldehyde, (**B**) paraformaldehyde, (**C**) methyl alcohol and (**D**) glutaraldehyde at 37 °C (**A,C**) or 4 °C (**B,D**) for one or 12 h. The fixatives were diluted with PBS at a fourfold gradient at 25 °C and used at respective temperatures for treatment. The detection methods of pre-F and total-F for these fixed samples refer to Figure 1. For statistical assay, a non-parametric statistical test was performed with GraphPad Prism version 7.00. Result means geometric mean ±95% CI. * *p* < 0.05, ** *p* < 0.01, *** *p* < 0.001, **** *p* < 0.0001. In addition, with samples at 1 h as references, the relative positive rate of samples at 12 h were compared. # *p* < 0.05, ## *p* < 0.01, ### *p* < 0.001, #### *p* < 0.0001.

**Figure 3 viruses-11-00628-f003:**
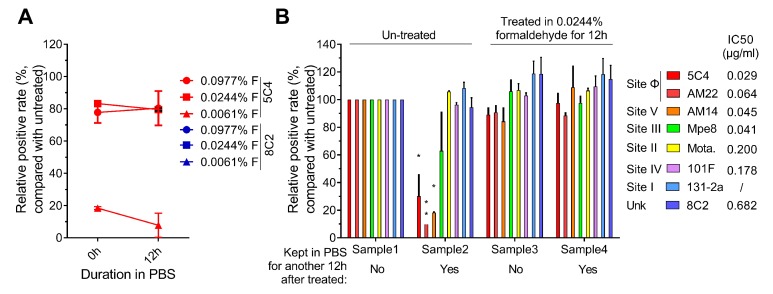
Epitope levels on fixed infected cells. (**A**) The reactive stability of fixed epitopes. HEp2 cells were infected with an RSV A2 strain, harvested by trypsin treatment at 24 h post-infection, and then were re-suspended in a 0.0061%, 0.0244% or 0.0977% formaldehyde solution for 12 h. The cells were then washed and re-suspended in PBS and incubated for another 0, 1 or 12 h at 37 °C. Then the cells were stained with 5C4 (red boxes) or 8C2 (blue boxes) and a secondary mAb conjugated with FITC and then detected by flow cytometry. (**B**) The presence of functional epitopes before and after incubating in PBS. HEp2 cells were infected, harvested and fixed as described in (**A**). After treatment with specific conditions, cells were stained with pre-F specific mAbs (5C4, AM22, AM14; red, light red, blue, respectively) or total F target mAbs (MPE8, MOTA, 8C2 or 101F; yellow, green, purple, orange, respectively) and a secondary mAb conjugated with FITC. The cells were then detected by flow cytometry. Samples 1 and 2 were not fixed with formaldehyde. These samples differ in that the cells in sample 1 were stained immediately after trypsin digestion, whereas the cells in sample 2 were incubated in PBS for 12 h prior to staining. Samples 3 and 4 were both treated with 0.0244% formaldehyde solution for 12 h; however, sample 4, but not sample 3, was further incubated in PBS for an additional 12 h prior to staining. With sample 1 as a reference, the relative positive rate of antigenic sites of other samples was compared. In (**A**) and (**B**), the positive rate of cells that reacted with the mAbs were measured and compared with untreated RSV-infected cells (the positive rate of this group was set as 100%). For statistical assay, a non-parametric statistical test was performed with GraphPad Prism version 7.00. Result means geometric mean ±95% CI. * *p* < 0.05, ** *p* < 0.01, *** *p* < 0.001, **** *p* < 0.0001.

**Figure 4 viruses-11-00628-f004:**
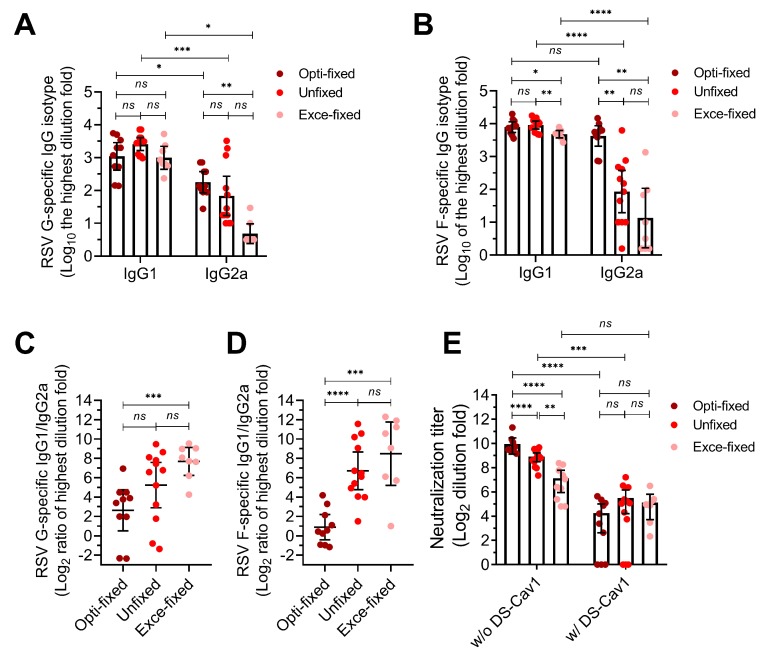
Serum antibodies response of immunized mice. Six-week-old BALB/c mice were inoculated with 1 × 10^7^ fixed cells per mouse at 0, 10, 20 and 30 days. Blood was collected for analysis in antibody isotype assay and neutralizing assay. (**A–D**) The G-specific and F-specific antibody titer. (**E**) The serum neutralizing activity with (w/) and without (w/o) adsorbing by DS-Cav1. MC means mock challenge (grey symbols); opti-fixed refers to mice inoculated with 0.0244% formaldehyde-treated infected cells (optimal concentration, dark red symbols); unfixed refers to mice inoculated with PBS-treated infected cells (red symbols); exce-fixed refers to mice inoculated with 25% formaldehyde-treated infected cells (excessively fixed, pink symbols). For statistical assay, two-tailed Mann-Whitney (nonparametric) tests were performed with GraphPad Prism version 7.00 to compare two groups. Result means geometric mean ±95% CI. * *p* < 0.05, ** *p* < 0.01, *** *p* < 0.001, **** *p* < 0.0001.

**Figure 5 viruses-11-00628-f005:**
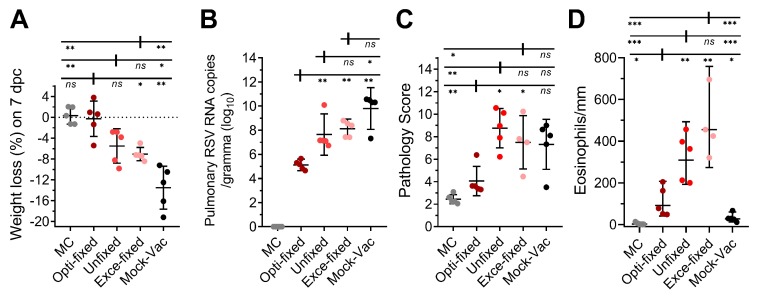
RSV challenge in BALB/c mice vaccinated with fixed cells. Six-week-old BALB/c mice were inoculated with 1 × 10^7^ fixed cells per mouse at 0, 10, 20 and 30 days. Blood was collected prior to inoculation for analysis in a neutralizing assay. (**A**) On day 58, mice were challenged with RSV A2 strain intranasally at a titer of 1 × 10^7^ PFU per mouse. Weight loss on day 7 post-challenge was analyzed. (**B**) Five days post-challenge, five mice from each group were sacrificed, and the right lungs were collected for analysis of pulmonary RSV via qPCR. (**C**) The left lungs were stored in 10% formaldehyde for pathological evaluation. (**D**) Perivascular eosinophils in the lung of mice on day 5 post challenge of RSV A2, eosinophils around the blood vessels were counted and the lengths of vessel walls were measured with micrometer. MC means mock challenge (grey symbols); opti-fixed refers to mice inoculated with 0.0244% formaldehyde-treated infected cells (optimal concentration, dark red symbols); unfixed refers to mice inoculated with PBS-treated infected cells (red symbols); exce-fixed refers to mice inoculated with 25% formaldehyde-treated infected cells (excessively fixed, pink symbols); mock-vac refers to mice inoculated with an equal quantity of uninfected cells. For statistical assay, two-tailed Mann-Whitney (nonparametric) tests were performed with GraphPad Prism version 7.00 to compare two groups. Result means geometric mean ±95% CI. * *p* < 0.05, ** *p* < 0.01, *** *p* < 0.001, **** *p* < 0.0001.

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
