# Peer review of "The Optimal Concentration of Formaldehyde is Key to Stabilizing the Pre-Fusion Conformation of Respiratory Syncytial Virus Fusion Protein"

_viruses, 2019, doi:10.3390/v11070628_

Round 1
Reviewer 1 Report
This is an interesting study which, if further validated, may present a novel approach to development of a safe, effective RSV vaccine. The authors are extending previous observations that the formalin-inactivated (FI-RSV) vaccine preparation from the 1960s did not contain adequate amounts of the immunogenic, pre-fusion conformation of RSV F, and the authors set out to determine conditions by which pre-fusion RSV F could be optimally preserved by different fixation conditions. The main findings are that if there is too low a concentration of formaldehyde (or paraformaldehyde), then pre-fusion RSV F is lost ndn the vaccine preparation is ineffective; moreover, if there is too high a concentration of fixative, then antigenic sites of pre-fusion RSV F are obscured through excessive cross-linking, rendering the vaccine preparation ineffective. The experiments show an optimal range of formaldehyde concentrations that can elicit high titers of RSV neutralizing antibodies and effective protection against RSV lung infection in a mouse model.
My comments are as follows:
One of the important aspects of FI-RSV vaccine enhanced pathology is an exaggerated Th2 response to RSV challenge. The authors should measure levels of Th-1 and Th2-related cytokines in the mouse model, to ensure their "opti-fixed" vaccine preparation does not elicit an exaggerated Th2 response. Knowing Th1/Th2 levels and balance would link the current work to previous literature on the topic.
The histological scoring system used (ref. 15) was designed for Mycoplasma pneumoniae infections, not RSV infections. The authors should redo histological scoring using a published method specifically designed for evaluation of vaccine-enhanced RSV lung histopathology. RSV vaccine-enhanced histopathology is different than that of RSV infection only, eg., increased eosinophils in lung inflammatory infiltrates.
In Figures 2 and 3, the bar graphs do not include error bars, so that the variability of the observations being reported are not known. These experiments should include several replicates and statistical comparisons between different experimental conditions.
In Figure 3B, the authors use antibodies to RSV F protein antigenic sites 0, V, III, II and IV. It is puzzling that antibodies to site I were not used (to finish the set of antigenic sites). This apparent omission requires explanation.
In Figure 2, the results for paraformaldehyde appear to be as good, or perhaps better, than those for formaldehyde. The authors should better explain their rationale with proceeding with a formalin-based, rather than a paraformaldehyde-based, vaccine for the mouse in vivo experiments.
Author Response
Point 1: One of the important aspects of FI-RSV vaccine enhanced pathology is an exaggerated Th2 response to RSV challenge. The authors should measure levels of Th-1 and Th2-related cytokines in the mouse model, to ensure their "opti-fixed" vaccine preparation does not elicit an exaggerated Th2 response. Knowing Th1/Th2 levels and balance would link the current work to previous literature on the topic.
Response 1: We agree the reviewer’s views on this. We measured the serum IgG antibody isotypes in mice that underwent three boost immunizations, and attempted to analyze the type of immunogen-triggered immune response using the ratio of IgG1 to IgG2a. The results showed that compared with unfixed group and exce-fixed group, Opti-fixed group mice were elicted to generate a significantly higher Th1 immune response, and Th1/Th2 in this group was balanced.
The serum G and F specific IgG antibodies analysis was showed in Line 297 to Line 308 as the newly added Figure-4A to 4D. And the corresponding description is from Line 284 to Line 290.
Point 2: The histological scoring system used (ref. 15) was designed for Mycoplasma pneumoniae infections, not RSV infections. The authors should redo histological scoring using a published method specifically designed for evaluation of vaccine-enhanced RSV lung histopathology. RSV vaccine-enhanced histopathology is different than that of RSV infection only, eg., increased eosinophils in lung inflammatory infiltrates.
Response 2: We thank the reviewer for the comment. The histological scoring system used was originally designed for the evaluation of acute Mycoplasma pneumonia, but has gradually been widely used in many studies for the evaluation of pulmonary inflammation caused by RSV infection [1-7].
Following your instructions, we additionally evaluated the increased eosinophils in lung inflammatory infiltrates. The newly added Figure is in Figure 5D, and the corresponding description is in Lines 340 to Line 343.
Point 3: In Figures 2 and 3, the bar graphs do not include error bars, so that the variability of the observations being reported are not known. These experiments should include several replicates and statistical comparisons between different experimental conditions.
Response 3: Thanks for the reviewer’s comment. According to your advice, we repeated the experiments and the statistical analysis was performed among different experimental groups. The modified Figure 2 and Figure 3 are at Line 219 and Line 250.
Point 4: In Figure 3B, the authors use antibodies to RSV F protein antigenic sites 0, V, III, II and IV. It is puzzling that antibodies to site I were not used (to finish the set of antigenic sites). This apparent omission requires explanation.
Response 4: We appreciate the reviewer’s comment. When we did this part of the experiment, we did not get 131-2A, so we did not use the antibody. Now we have purchased this antibody and supplemented the data with it, as shown in modified Figure 3 at Line 250.
Point 5: In Figure 2, the results for paraformaldehyde appear to be as good, or perhaps better, than those for formaldehyde. The authors should better explain their rationale with proceeding with a formalin-based, rather than a paraformaldehyde-based, vaccine for the mouse in vivo experiments.
Response 5: We appreciate the reviewer’s comment. Paraformaldehyde is not stable, easy to depolymerize, in general, it was used in 4°C, and at the moment, our results (data not shown in the manuscript) show that under different temperatures it’s fixed effect is varying (as shown below), which is not suitable for technologize. So we just used the formaldehyde for further assay.
1. Numata, M., P. Kandasamy, Y. Nagashima, R. Fickes, R.C. Murphy,D.R. Voelker, Phosphatidylinositol inhibits respiratory syncytial virus infection. J Lipid Res, 2015. 56(3): p. 578-87 DOI: 10.1194/jlr.M055723.
2. Lu, X., K.S. McCoy, J. Xu, et al., Galectin-9 ameliorates respiratory syncytial virus-induced pulmonary immunopathology through regulating the balance between Th17 and regulatory T cells. Virus Res, 2015. 195: p. 162-71 DOI: 10.1016/j.virusres.2014.10.011.
3. Alsuwaidi, A.R., A. Albawardi, S. Almarzooqi, et al., Respiratory syncytial virus increases lung cellular bioenergetics in neonatal C57BL/6 mice. Virology, 2014. 454-455: p. 263-9 DOI: 10.1016/j.virol.2014.02.028.
4. Numata, M., Y. Nagashima, M.L. Moore, et al., Phosphatidylglycerol provides short-term prophylaxis against respiratory syncytial virus infection. J Lipid Res, 2013. 54(8): p. 2133-43 DOI: 10.1194/jlr.M037077.
5. Numata, M., Y.V. Grinkova, J.R. Mitchell, H.W. Chu, S.G. Sligar,D.R. Voelker, Nanodiscs as a therapeutic delivery agent: inhibition of respiratory syncytial virus infection in the lung. Int J Nanomedicine, 2013. 8: p. 1417-27 DOI: 10.2147/ijn.S39888.
6. Dou, Y., Y. Zhao, Z.Y. Zhang, H.W. Mao, W.W. Tu,X.D. Zhao, Respiratory syncytial virus infection induces higher Toll-like receptor-3 expression and TNF-alpha production than human metapneumovirus infection. PLoS One, 2013. 8(9): p. e73488 DOI: 10.1371/journal.pone.0073488.
7. Numata, M., H.W. Chu, A. Dakhama,D.R. Voelker, Pulmonary surfactant phosphatidylglycerol inhibits respiratory syncytial virus-induced inflammation and infection. Proc Natl Acad Sci U S A, 2010. 107(1): p. 320-5 DOI: 10.1073/pnas.0909361107.

Reviewer 2 Report
This manuscript by Zhang et al, examines whether optimization of basic chemical fixatives can stabilize the prefusion F conformation, and thereby pave the way for future simple preparation of safe and effective inactivated RSV vaccine. The data show that by vaccinating mice with RSV infected cells, fixed using different concentrations of (para) formaldehyde, neutralizing antibody titers and the disease outcome can be improved. Whereas the data are interesting, worthwhile, and show promise through improvements in pathogenesis outcome, a number of questions on the methods/platform used (infected whole cells as vaccines), and potential contributions of G and SH which are not addressed, cast some doubt as to the strengths and applicability of the findings and conclusions.
Major concerns
Can the authors exclude that the improvements seen in the disease outcome in mice are due, or partly due, to impact of fixation on G (and/or SH), and not of F protein stabilization? This could be shown by neutralization of a virus deleted in G, which are available in the field.
Rather than looking at the effect of fixation on virions, the authors use trypsinized RSV infected cells to vaccinate. This is not necessarily comparable to vaccines based on purified virions and would not be an acceptable method of vaccination as a relatively high degree of vaccine purity will be required. Presence of cells in the vaccine could also have adjuvanting properties that could influence the animal outcomes, especially after repeated vaccinations (4 doses) used here. Thus it is not clear whether the findings translate well to practice, as purification of virions would lead to conformationally shifted F before fixation can even begin. In the discussion, the authors indicate they did also examine fixation of virions. This would be much more relevant than looking at infected cells, yet the virion data are listed ‘not shown’.
The infected cells used for the experiments were harvested after trypsin treatment and kept in a buffer (PBS) not optimal for live cells. Trypsin treatment could cleave or otherwise impact the surface proteins including F and G, and keeping cells in PBS for extended periods also has impact the cells and could negatively affect F stability of unfixed samples, which are the control to which fixation is compared. The concentration of F on the surface of infected cells likely differs from that of F protein embedded in the virion membrane, and this may also impact conformation stability, for example by the extent of contact with matrix protein. In short, using harvested infected cells and keeping unfixed controls for extended periods in PBS, creates some unknowns that make the data less convincing.
Fig. 1 and 2 are based on only one pre-fusion Ab, even though there likely are in between conformations, and authors have access to additional antibodies. Since the main goal is to show that fixation does not cause any conformational abnormalities or changes, the use of one Ab is not adequate. In Fig. 3, two additional pre-fusion specific antibodies are used, which strengthens the data. Nevetheless, additional antibodies should be used in Fig 1 and 2.
In Fig 1 and 2, statistical significance is lacking. Since these are straightforward experiments, additional experiments or samples could be examined to determine whether the results are significant or not.
Minor items
- There are amino acid differences between F proteins from A2 strains from different labs that impact stability. Since only one F protein is tested, one cannot know whether the improvements will be applicable to other F proteins, even within the same strain.
- In Fig 1 and 2, every panel is identical, and it takes effort to find out what is what. The authors should add information to the panels (for example, temperature, which antibody), so the reader can more readily interpret the data.
- Authors say they use a ‘Cimolai’ method to evaluate histopathology. There should be a brief description of how the scoring is actually done. Also include an explanation what ‘quality of peribronchiolar/bronchial infiltrates’ is.
Author Response
Major comments
Point 1: Can the authors exclude that the improvements seen in the disease outcome in mice are due, or partly due, to impact of fixation on G (and/or SH), and not of F protein stabilization? This could be shown by neutralization of a virus deleted in G, which are available in the field.
Response 1: We thank the reviewer for the comment. In Line 302 to Line 305, and Figure-4E, we showed the neutralization contribution of non-F antibodies. Since we do not have viruses with G or SH deletion, we used recombinant pre-F trimer (DS-Cav1) adsorbed based neutralization assay to reflect the contribution of serum antibodies targeting non-F proteins (G or SH) to neutralization, according to the recently report [1]. As shown in the Figure 4E, after the serums were adsorbed with sufficient amount of DS-Cav1, their neutralization ability was almost eliminated. This indicated that, the major neutralizing antibody targeted F protein in the serum. However, we also found the antibodies targeting non-F protein in the serums of Unfix group, which had a certain neutralization effect (Figure-4, panel E).
Point 2: Rather than looking at the effect of fixation on virions, the authors use trypsinized RSV infected cells to vaccinate. This is not necessarily comparable to vaccines based on purified virions and would not be an acceptable method of vaccination as a relatively high degree of vaccine purity will be required. Presence of cells in the vaccine could also have adjuvanting properties that could influence the animal outcomes, especially after repeated vaccinations (4 doses) used here. Thus it is not clear whether the findings translate well to practice, as purification of virions would lead to conformationally shifted F before fixation can even begin. In the discussion, the authors indicate they did also examine fixation of virions. This would be much more relevant than looking at infected cells, yet the virion data are listed ‘not shown’.
Response 2: We thank the reviewer for the comment. We may not be making it clear enough here. We agree that there is a great risk of pre-F allostery in the production of RSV virions, but at present, we do not have a suitable process to obtain viruses with high pre-F levels. Under the current conditions, good protective effect can be achieved by using fixed cells to immunize mice, which shows the potential of this fixative condition for inactivated RSV vaccine. Now we are also exploring inactivated vaccines based on virions, and preliminary results indicated that the optimal conditions obtained by cell screening could also be applied to viruses in certain degree (Figure S3). We appended the method to detect F protein on virions in Line 160 to Line 170.
As for the adjuvant properties of cells, the uninfected cells treated with fixative could not protect immunized mice (as shown in the following figure), indicating that the presence of cells failed to have a positive effect on the animal outcomes.
Point 3: The infected cells used for the experiments were harvested after trypsin treatment and kept in a buffer (PBS) not optimal for live cells. Trypsin treatment could cleave or otherwise impact the surface proteins including F and G, and keeping cells in PBS for extended periods also has impact the cells and could negatively affect F stability of unfixed samples, which are the control to which fixation is compared. The concentration of F on the surface of infected cells likely differs from that of F protein embedded in the virion membrane, and this may also impact conformation stability, for example by the extent of contact with matrix protein. In short, using harvested infected cells and keeping unfixed controls for extended periods in PBS, creates some unknowns that make the data less convincing.
Response 3: We thank the reviewer for the comment. We may not be making it clear enough in the manuscript. As shown in Figure 1-3, F protein after trypsin treatment still retained the major antigenic sites, which may indicate that the possible effect of trypsin treatment on F protein was small and existed in all groups, which was comparable among groups. We added the corresponding explanation in Line 372 to Line 373.
We also observed the storage of the unfixed cells in PBS, and found that there was no obvious clustering of cells within 12 hours, and no obvious deviation of cell population was detected by flow method. Meanwhile, total-f (8C2 reaction value) was relatively stable.
Point 4: Fig. 1 and 2 are based on only one pre-fusion Ab, even though there likely are in between conformations, and authors have access to additional antibodies. Since the main goal is to show that fixation does not cause any conformational abnormalities or changes, the use of one Ab is not adequate. In Fig. 3, two additional pre-fusion specific antibodies are used, which strengthens the data. Nevetheless, additional antibodies should be used in Fig 1 and 2.
Response 4: We thank the reviewer for the comment. We may not be making it clear enough here. The early screening of fixative conditions requires a lot of antibody, but limited by incomplete eukaryotic expression system, we cannot easily get these antibodies by eukaryotic expression, so we only tested the reaction of 5C4 and 8C2 (these two antibodies generated by purification of ascites, it is easy). The results showed that the optimal fixative condition could stabilize the pre-F-specific site0, but as you said, it cannot directly represent pre-F. Therefore, for further evidence, we used rare eukaryotic antibodies to detect other epitopes of F protein with partial condition fixed, and the results showed that the main antigenic sites of pre-F protein were also present in Figure-3B. We added the corresponding explanation in Line 178 to Line 180.
Point 5: In Fig 1 and 2, statistical significance is lacking. Since these are straightforward experiments, additional experiments or samples could be examined to determine whether the results are significant or not.
Response 5: We thank the reviewer for the comment. Supplementary experiments have been conducted according to your Suggestions, and corresponding modifications have been made in Figure 1 and Figure 2.
Minor comments
Point 1: There are amino acid differences between F proteins from A2 strains from different labs that impact stability. Since only one F protein is tested, one cannot know whether the improvements will be applicable to other F proteins, even within the same strain.
Response 1: We thank the reviewer for the comment. Formaldehyde is very easy to react with protein molecules. Formaldehyde at the appropriate concentration can form an appropriate amount of chemical cross-linking between the amino acids of the pre-F protein, such as methylene, which may prevent the rearrangement allostery within the pre-F protein molecule. This concentration-dependent modification law is universal to some extent. We tried to fix F proteins with different stable properties on the virus (not shown), and the results were consistent.
Point 2: In Fig 1 and 2, every panel is identical, and it takes effort to find out what is what. The authors should add information to the panels (for example, temperature, which antibody), so the reader can more readily interpret the data.
Response 2: We thank the reviewer for the suggestion. It has been modified according to your suggestion in Figure 1 and Figure 2.
Point 2: Authors say they use a ‘Cimolai’ method to evaluate histopathology. There should be a brief description of how the scoring is actually done. Also include an explanation what ‘quality of peribronchiolar/bronchial infiltrates’ is.
Response 2: We thank the reviewer for the suggestion. As you suggested, we added the description of Cimolai rule in the Materials and Method in Line 119 to Line 125.
‘Quality of peribronchiolar/bronchial infiltrates’ describes the degree of infiltration of peribronchiolar inflammatory cells, which can be divided into four grades. ‘0’ represents no peribronchiolar inflammatory cell aggregation; ‘1’ represents a mild or slight infiltration, often with interrupted collar; ‘2’ represents a moderate infiltration, complete collar or crescent collar with<5 cells="" represents="" a="" severe="" complete="" collar="" with="">5-10 cells thickness.
1. Ngwuta, J.O., M. Chen, K. Modjarrad, et al., Prefusion F-specific antibodies determine the magnitude of RSV neutralizing activity in human sera. Sci Transl Med, 2015. 7(309): p. 309ra162 DOI: 10.1126/scitranslmed.aac4241.

Round 2
Reviewer 1 Report
In general, the revised manuscript is much improved over the original submission but in my opinion there are still a number of things that have to be addressed (most very minor and one major) before it can be published in Viruses:
Line 27: should read: “BALB/c mice”
Line 54: Should read: “phase II clinical trials”
Line 118: Revise to read: “A pathologist who did not have knowledge of the
group assignment of individual mice scored lung pathology according to the Cimolai method [15].”
Line 155: Should read: “Sera were then serially 10-fold diluted”
Line 181: Should read: “A p value of<0.05 was considered statistically significant.”
Lines 207, 237 and 281: Should use a non-parametric statistical test rather than unpaired t-tests.
Line 208, 237, 282, 320 and 377: I do not understand “Result means genomic mean +/- 95% CI.” Is it the geometric mean? If so, please correct.
Line 236: Change to: “pre-F” and “total F”
Lines 297-303: The authors apparently did not understand my comment in the initial review regarding Th1 and Th2 cytokines and Th1/Th2 balance. For assessment of Th1 and Th2 responses and Th1/Th2 balance, levels of cytokines (mRNA or protein) such as IL-2 and IFN-gamma (Th1-markers) and IL-4, IL-5 and IL-13 (Th2-markers) should be quantified in lung samples. Circulating IgG1 and IgG2a antibody titers in the blood, while interesting, are not adequate to address the issues related to mechanisms of enhanced lung pathology of RSV-infected mice that received formalin-inactivated vaccine (i.e., Th2-mediated). If it is not possible to measure lung Th1- and Th2-cytokines in the lungs, the authors should comment on the increase of lung eosinophils in the opti-fixed group compared to the MC group as this may reflect a Th2-shift in Th1/Th2 balance, noting that this happened to a significantly lesser extent than what was observed the unfixed or exce-fixed groups.
Lines 339-344. Rewrite this to read, “Results of lung pathology scoring are shown in Figure 5C and Figure S1. The mock-vaccinated mice developed relatively serious pulmonary inflammation and slight perivascular inflammation, with the appearance of infiltrating cells in the thickened alveolar wall and alveolar space (Figure S2).”
Author Response
Response to Reviewer 1 Comments
Point 1: Line 27: should read: “BALB/c mice”.
Response 1: We thank the reviewer for the comment. It has been modified according to your suggestion in Line 27.
Point 2: Line 54: Should read: “phase II clinical trials”.
Response 2: We thank the reviewer for the comment. It has been modified according to your suggestion in Line 57.
Point 3: Line 118: Revise to read: “A pathologist who did not have knowledge of the group assignment of individual mice scored lung pathology according to the Cimolai method [15].
Response 3: Thanks for the reviewer’s comment. It has been modified according to your suggestion in Line 123.
Point 4: Line 155: Should read: “Sera were then serially 10-fold diluted”.
Response 4: We appreciate the reviewer’s comment. It has been modified according to your suggestion in Line 158-159.
Point 5: Line 181: Should read: “A p value of<0.05 was considered statistically significant.”
Response 5: We appreciate the reviewer’s comment. It has been modified according to your suggestion in Line 184.
Point 6: Lines 207, 237 and 281: Should use a non-parametric statistical test rather than unpaired t-tests.
Response 6: We appreciate the reviewer’s comment. They have been modified according to your suggestion in Lines 210-211, 240-241 and 284-285.
Point 7: Line 208, 237, 282, 320 and 377: I do not understand “Result means genomic mean +/- 95% CI.” Is it the geometric mean? If so, please correct.
Response 7: We appreciate the reviewer’s comment. It is the geometric mean. They have been modified according to your suggestion in Lines 211-212, 241, 285, 323 and 385.
Point 8: Line 236: Change to: “pre-F” and “total F”
Response 8: We appreciate the reviewer’s comment. It has been modified according to your suggestion in Line 239.
Point 9: Lines 297-303: The authors apparently did not understand my comment in the initial review regarding Th1 and Th2 cytokines and Th1/Th2 balance. For assessment of Th1 and Th2 responses and Th1/Th2 balance, levels of cytokines (mRNA or protein) such as IL-2 and IFN-gamma (Th1-markers) and IL-4, IL-5 and IL-13 (Th2-markers) should be quantified in lung samples. Circulating IgG1 and IgG2a antibody titers in the blood, while interesting, are not adequate to address the issues related to mechanisms of enhanced lung pathology of RSV-infected mice that received formalin-inactivated vaccine (i.e., Th2-mediated). If it is not possible to measure lung Th1- and Th2-cytokines in the lungs, the authors should comment on the increase of lung eosinophils in the opti-fixed group compared to the MC group as this may reflect a Th2-shift in Th1/Th2 balance, noting that this happened to a significantly lesser extent than what was observed the unfixed or exce-fixed groups.
Response 9: We appreciate the reviewer’s comment. We are very sorry for misunderstanding the reviewer's comments. We agree with you that cytokines in the lung samples are more directly responsible for this problem. In fact, we tried to test for cytokines in mice lung homogenates previously stored in the refrigerator at - 80 °C, but we found that the cytokines in these samples became undetectable, possibly because the samples underwent three times of freezing and thawing. The antibodies in the serum were relatively stable, so we also examined the antibody isotypes in the serum to try to show the effect of the immunogen on the Th1/Th2 balance, as reported in some articles [1, 2]. The results showed that the Th1/Th2 balance reflected by the serum antibody isotypes was consistent with the increase of pulmonary eosinophils and pulmonary pathology score, which was interesting, so we included it in the manuscript.
Accordingly, we followed your advice and added a comment on the increase of pulmonary eosinophils and their possible effects in line 359-364.
Point 10: Lines 339-344. Rewrite this to read, “Results of lung pathology scoring are shown in Figure 5C and Figure S1. The mock-vaccinated mice developed relatively serious pulmonary inflammation and slight perivascular inflammation, with the appearance of infiltrating cells in the thickened alveolar wall and alveolar space (Figure S2).”
Response 10: We appreciate the reviewer’s comment. It has been modified according to your suggestion in Line 343-346.
1. Bayer, L., J. Fertey, S. Ulbert,T. Grunwald, Immunization with an adjuvanted low-energy electron irradiation inactivated respiratory syncytial virus vaccine shows immunoprotective activity in mice. Vaccine, 2018. 36(12): p. 1561-1569 DOI: 10.1016/j.vaccine.2018.02.014.
2. Lee, Y., E.J. Ko, K.H. Kim, et al., The efficacy of inactivated split respiratory syncytial virus as a vaccine candidate and the effects of novel combination adjuvants. Antiviral Res, 2019. 168: p. 100-108 DOI: 10.1016/j.antiviral.2019.05.011.

Reviewer 2 Report
In this re-submitted manuscript of W. Zhang et al, the authors have addressed most concerns stated in the first review. A number of arguments have been added to strengthen or explain the results, figures have received additional labels, statistical significance was added as well as an entirely new Figure (Fig 4). Whereas many points have been properly addressed, the 1st and 2nd major concerns were not (see below), and as a result the strengths of the current findings remain limited.
Questions regarding the contribution of F. Fig 4 is a new figure in the re-submitted manuscript to address previous reviewer concerns, in which panel E is supposed to demonstrate that protection is mostly F-mediated. However the used technique is taken from reference 16 (Ngwuta et al) which was specifically designed to adsorb F antibodies from human sera. This technique does not work for mice sera because the sheep anti-mouse IgG beads (in being anti-mouse) will not only remove prefusion F antibodies via strep-tag II (mouse-based) antibodies but will also directly bind and remove all other (non prefusion F) mouse IgG from the sera. This reduces both prefusion F -based and non-prefusion F -based neutralization, invalidating this figure and indicating that the drop in neutralization may or may not be specific to F.
A second complication pertaining to the same problem, is that in vitro neutralization by G antibodies in general does not work well, because G has heparin binding domains which bind non-specifically but strongly to heparin sulfate on almost all cultured cells. This means that absence of in vitro neutralization by G antibodies does not indicate absence of in vivo neutralization by G antibodies.
Together, it is far from clear that protection (and benefits from the fixation method) is largely mediated through F (lines 306-308 and 413), and thus the effects of fixation could also be mediated by other (G, SH) proteins. Either a more appropriate method needs to be employed to show that the benefit of fixation is F-based, or all results and conclusions pertaining to the role of F should be modified.
Usefulness of whole-cell vaccination versus purified virus vaccination. This concern is not major, but does somewhat limit the impact of Figures 4 and 5, and the paper overall. It is now well published and established that stabilization of preF induces more efficient protection from RSV. If the goal of this study is to indicate potential for simple stabilization of preF by fixation, then an antibody-recognition based study (Figs 1-3) would suffice, as there are many conformation-specific antibodies that have been validated. Figs 4 and 5 (vaccinating with fixed whole infected cells) confirm the already known benefits of preF stabilization, but do not provide much additional insight, as whole cell vaccination is not a realistic vaccine approach and is not comparable to purified virus based vaccines. The adjuvant question was not adequately addressed: Absence of protection by uninfected cells (mock vaccination) does not exclude the fact that cells can act as adjuvant in the context of viral antigens. Figures 1-3 do demonstrate that prefusion F can be stabilized on the surface of infected cells and this is the main finding with value for future purified inactivated virus vaccines. Figures 4 and 5 could remain in the manuscript, but unless Fig 4E is improved, they have only moderate value.
Other concerns
In response to concerns, a supplemental figure (S3) was added showing fixation of purified virions. However, the figure is unclear, lacks controls, and does not add any conclusive data and is not helpful to the manuscript.
Author Response
Response to Reviewer 2 Comments
Point 1: Questions regarding the contribution of F. Fig 4 is a new figure in the re-submitted manuscript to address previous reviewer concerns, in which panel E is supposed to demonstrate that protection is mostly F-mediated. However the used technique is taken from reference 16 (Ngwuta et al) which was specifically designed to adsorb F antibodies from human sera. This technique does not work for mice sera because the sheep anti-mouse IgG beads (in being anti-mouse) will not only remove prefusion F antibodies via strep-tag II (mouse-based) antibodies but will also directly bind and remove all other (non prefusion F) mouse IgG from the sera. This reduces both prefusion F -based and non-prefusion F -based neutralization, invalidating this figure and indicating that the drop in neutralization may or may not be specific to F.
A second complication pertaining to the same problem, is that in vitro neutralization by G antibodies in general does not work well, because G has heparin binding domains which bind non-specifically but strongly to heparin sulfate on almost all cultured cells. This means that absence of in vitro neutralization by G antibodies does not indicate absence of in vivo neutralization by G antibodies.
Together, it is far from clear that protection (and benefits from the fixation method) is largely mediated through F (lines 306-308 and 413), and thus the effects of fixation could also be mediated by other (G, SH) proteins. Either a more appropriate method needs to be employed to show that the benefit of fixation is F-based, or all results and conclusions pertaining to the role of F should be modified.
Response 1: We thank the reviewer for the comment. We quite agree with you on the contribution of F. The complexity of the immunogen determines the complex composition of the antibodies it stimulates. We recognized the flaw in the method of serum adsorption assay you mentioned, and then we tried the competitive neutralization of F protein to analyze the contribution of F antibody to neutralization. In other words, sufficient amount of F was pre-incubated with serum to bind anti-F antibody in serum, and then the mixture was used to neutralize RSV virus. We found that the addition of F protein eliminated the main neutralization ability in serum, indicating that the contribution of anti-f antibody to neutralization was important. However, as you pointed out, anti-G antibody neutralization is difficult to be accurately evaluated in vitro, except with human respiratory epithelium (HAE) model, so we also believe that G or SH antibodies may contribute to serum neutralization.
We have modified the description of methods and results accordingly, see Line 139-141, 308-311, 325, 421-423, 428.
Point 2: Usefulness of whole-cell vaccination versus purified virus vaccination. This concern is not major, but does somewhat limit the impact of Figures 4 and 5, and the paper overall. It is now well published and established that stabilization of preF induces more efficient protection from RSV. If the goal of this study is to indicate potential for simple stabilization of preF by fixation, then an antibody-recognition based study (Figs 1-3) would suffice, as there are many conformation-specific antibodies that have been validated. Figs 4 and 5 (vaccinating with fixed whole infected cells) confirm the already known benefits of preF stabilization, but do not provide much additional insight, as whole cell vaccination is not a realistic vaccine approach and is not comparable to purified virus based vaccines. The adjuvant question was not adequately addressed: Absence of protection by uninfected cells (mock vaccination) does not exclude the fact that cells can act as adjuvant in the context of viral antigens. Figures 1-3 do demonstrate that prefusion F can be stabilized on the surface of infected cells and this is the main finding with value for future purified inactivated virus vaccines. Figures 4 and 5 could remain in the manuscript, but unless Fig 4E is improved, they have only moderate value.
Response 2: We thank the reviewer for the comment. We agree with you that cell vaccination is not really a realistic vaccine approach, which is why we continue to explore pre-f fixation on viruses. Indeed, previous reports have confirmed that pre-f induces a better neutralizing antibody response, but a key question in this paper is whether the formaldehyde-treated F protein could be presented to the immune system in vivo as pre-f. The results showed that immunogen treated with appropriate concentrations of formaldehyde could induce higher serum neutralization activity, which indirectly proved that stable pre-F could be relatively stable in vivo.
In addition, in accordance with your previous suggestion, we reanalyzed the contribution of F antibody to neutralization by using competitive neutralization based on F protein, and made corresponding modifications to Fig 4E.
Point 3: In response to concerns, a supplemental figure (S3) was added showing fixation of purified virions. However, the figure is unclear, lacks controls, and does not add any conclusive data and is not helpful to the manuscript.
Response 3: We thank the reviewer for the comment. We apologize for not showing enough of our data on virions in our previous response. Now we have perfected the supplementary Figure-S3 according to your suggestion, and added panel B in the figure to display the quantization result of signal of panel A.

Round 3
Reviewer 2 Report
In this re-submitted manuscript of W. Zhang et al, the authors have adequately addressed most concerns stated in the previous review. A different method was employed to ascertain the contribution of anti prefusion F antibodies to virus neutralization, and the possibility of a contribution by anti G and SH antibodies has been added in the text. Although the data support improved stabilization of the prefusion F conformation by fixation, it should be kept in mind that the 'stabilized' vaccine consists of infected cells, not virus.
Two minor concerns remain:
There is no mention in the discussion that the ‘immunogen’ consisted of infected cells, not virus. This should be acknowledged in the discussion, also because the virus data in supplemental Fig 3 are preliminary and not conclusive.
Supplemental Fig 3: standard deviation requires triplicate values, whereas only duplicate values are shown, without further information.
Author Response
Response to Reviewer 2 Comments
Point 1: There is no mention in the discussion that the ‘immunogen’ consisted of infected cells, not virus. This should be acknowledged in the discussion, also because the virus data in supplemental Fig 3 are preliminary and not conclusive.
Response 1: We thank the reviewer for the comment. According to your suggestion, we have made corresponding modifications in Line 409 to Line 411.
Point 2: Supplemental Fig 3: standard deviation requires triplicate values, whereas only duplicate values are shown, without further information.
Response 2: We thank the reviewer for the comment. We apologize for not adequately describing the results. In fact, supplementary Fig 3 represents three independent repeated experiments, panel A is a representative representation of these three independent repeated experiments, and panel B refers to the geometric mean of the three independent repeated experiments and 95% CI.
In addition, to avoid confusion, we have supplemented the detailed description in the corresponding legend based on your comments.
